# Neutrophil Extracellular Traps in Colorectal Cancer Progression and Metastasis

**DOI:** 10.3390/ijms22147260

**Published:** 2021-07-06

**Authors:** Umama Khan, Sabrina Chowdhury, Md Morsaline Billah, Kazi Mohammed Didarul Islam, Henrik Thorlacius, Milladur Rahman

**Affiliations:** 1Biotechnology and Genetic Engineering Discipline, Khulna University, Khulna 9208, Bangladesh; umamakhan140717@gmail.com (U.K.); morsaline@bge.ku.ac.bd (M.M.B.); didar950718@yahoo.com (K.M.D.I.); 2Biochemistry and Biotechnology, North South University, Dhaka 1229, Bangladesh; sabrinaachowdhury6@gmail.com; 3Department of Clinical Sciences, Malmö, Section for Surgery, Lund University, 214 28 Malmö, Sweden; henrik.thorlacius@med.lu.se

**Keywords:** colorectal cancer, metastasis, neutrophil, extracellular DNA, therapeutics

## Abstract

Neutrophils form sticky web-like structures known as neutrophil extracellular traps (NETs) as part of innate immune response. NETs are decondensed extracellular chromatin filaments comprising nuclear and cytoplasmic proteins. NETs have been implicated in many gastrointestinal diseases including colorectal cancer (CRC). However, the regulatory mechanisms of NET formation and potential pharmacological inhibitors in the context of CRC have not been thoroughly discussed. In this review, we intend to highlight roles of NETs in CRC progression and metastasis as well as the potential of targeting NETs during colon cancer therapy.

## 1. Introduction

Colorectal cancer (CRC) is the second most common cause of cancer-related deaths in the western world [1]. Every year, the management of CRC imposes a huge economic burden towards the health care system in Europe and other countries in the world [2]. CRC originates from benign, precancerous proliferative growth known as polyps [3]. During the slow development phases of polyp, different mutations start to accumulate and transform some of the polyps into malignant carcinoma [3]. Early stages of CRC are curable by surgery, however, when cancer metastasizes to lymph nodes or other distant organs, the prognosis of CRC becomes poor [4]. Furthermore, about 20% of CRC patients have already progressed into a metastatic state at the time of presentation and more than 30% of patients with early CRC have been reported to develop metastatic disease eventually [5,6]. The most common site of CRC metastasis is found to be the liver (about 70% of patients), followed by lung, distant lymph nodes, and peritoneum [7]. Once CRC is metastasized to multiple organs, its treatment becomes palliative rather than curative. During metastasis, cancer cells express certain characteristics, which include elevated expression of cell adhesion molecules, chemokine receptors, and increased cytoskeletal changes to favor migration in response to chemotactic signals to distant organs [8,9,10].

Accumulating studies suggest that approximately 25% of all tumors originates from chronic inflammation [11,12,13]. It is possible that inflammation can generate numerous growth factors and chemo-attractants to promote cancer cell proliferation, adhesion, and migration. During innate immune response, neutrophils play a key role by engulfing the invading pathogens directly or releasing antimicrobial agents to kill them. Interestingly, the increased accumulation of neutrophils was observed in pre-metastatic organs [14,15]. In addition to phagocytosis, neutrophils can form sticky web-like structures of decondensed chromatin filaments, decorated with histones and neutrophil granule proteins known as neutrophil extracellular traps (NETs) [16]. NETs have been reported to be involved in cancer development [17,18,19] and metastasis [20,21]. A study has revealed the association of NET formation inside the microvasculature by systemic inflammation with trapping of cancer cells in both liver and lung [22]. In addition, intravascular NET increases vascular permeability and promotes cancer cell extravasation from blood vessels to organs [23]. Interestingly, surgical stress facilitates cancer metastasis through associating it with inflammation [24,25]. Furthermore, immunostaining of tissue samples from CRC patients revealed the presence of NETs in primary tumor and associated metastatic lymph nodes [26]. Based on this evidence, it could be suggested that NETs might be involved in colon cancer cell proliferation and metastasis. This review will focus on the roles of NETs in CRC progression and metastasis, as well as the possibility of targeting NETs during cancer therapy.

## 2. Mechanism of NET Formation

The process in which NET formation occurs is known as NETosis. Initially, NETosis was referred to as the new type of defensive neutrophil death, however, later it was found that pathogenic stimulation could also induce vital and rapid production of NETs without effecting neutrophil viability [27]. Two types of mechanism have been proposed to elucidate NET formation: NADPH-oxidase (NOX)-dependent lytic NET formation and NADPH-oxidase (NOX)-independent non-lytic NET formation [28] (Figure 1).

### 2.1. NADPH-Oxidase (NOX)-Dependent Lytic NET Formation

The process of lytic NET formation initiates with the recognition of pathogens or sterile stimuli by various cellular receptors including toll-like receptors (TLRs), antibody fragment (Fc) receptors, complement receptors, etc. [27]. The initial activation via cellular receptors triggers calcium release from endoplasmic reticulum (ER), which in turn activates protein kinase C (PKC) and NADPH-oxidase complex, finally forming reactive oxygen species (ROS) [29]. In particular, ROS can activate protein-arginine deiminase 4 (PAD4), an enzyme that decondenses nuclear chromatin by converting arginine to citrulline [30]. Furthermore, neutrophil granular protein, myeloperoxidase facilitates neutrophil elastase translocation to the nucleus and unfolds chromatin [31]. This results in the breakdown of nuclear membrane and the release of decondensed chromatin into cytosol where released DNA is further decorated with granular and cytosolic proteins [32]. Finally, NETs are released through disruption of the plasma membrane, and when the neutrophil dies. Interestingly, some studies showed that mitochondrial DNA could also be expelled as NETs in response to inflammation [33,34].

### 2.2. NADPH-Oxidase (NOX)-Independent Non-Lytic NET Formation

Several studies have revealed that the formation of NETosis is independent of cell death [35,36]. The process of NET formation without cell death is known as non-lytic or vital NETosis which usually occurs in the absence of NADPH-oxidase pathway and does not lead to the formation of oxidants [27]. The major difference between lytic and non-lytic NETosis is that non-lytic NETosis takes place within minutes of stimulation without ROS formation while lytic NETosis needs several hours of stimulation and ROS formation. In non-lytic NET formation, neutrophils activation is induced by bacteria or bacterial products or activated platelets or complement proteins [27]. Chromatin de-condensation and neutrophil elastase translocation to the nucleus take place in a similar manner to lytic NET formation. However, chromatin decorated with cytosolic and nuclear proteins is discharged by the blebbing nuclear envelope rather than the overt breakdown of the nuclear membrane. Nuclear membrane blebbing and vesicle-mediated extracellular transport of NETs occur independent of plasma membrane disintegration [37,38].

## 3. NETs in Gastrointestinal Inflammation

NETs have been implicated in many gastrointestinal diseases including inflammatory bowel diseases (IBDs) [39,40,41], liver disease [35], and acute pancreatitis [42,43,44]. A chronic abiding repetitive event of inflammation in intestinal epithelium is portrayed as IBD [39]. Aberrant NET accumulation and the deterioration of inflamed intestinal barrier integrity can be correlated with IBD [39], as in ulcerative colitis (UC) [40]. Neutrophils and NET-associated molecules are found in abundance in both colonic biopsies of UC [45,46] and Crohn’s disease (CD) [41], which indicate intestinal inflammatory aggravation, epithelial exasperation, and elevated thrombotic impulse [47]. Administration of dextran sulfate sodium (DSS) [36,46] or 2,4,6-trinitrobenzene sulfonic acid (TNBS) [40] can induce murine UC, principally by disrupting intestinal epithelial barrier and eliciting a number of chronic immunologic responses. In the mouse model, increased DSS consumption can elevate the plasma level of extracellular DNA (ecDNA) in the form of NETs, which in turn induces UC [39]. The elevated response is correlated with the amplitude of disease severity and relative proportion of cells undergoing NETosis. However, the initial administration of protein-arginine deaminase type 4 (PAD4) inhibitors, Cl-amidine and streptonigrin can curb the formation of NETs, and in turn decrease the plasma ecDNA level in UC [39]. Some other studies have reported that the systemic administration of DNase can lead to amelioration of DSS-induced colitis by dissolving NETs [39,47]. In addition, the elevated level of NET-inducing protein, PAD4 is found in colon biopsies of active UC and CD cases compared to healthy patient samples [46]. The detrimental consequence of abnormal NET formation can lead to intestinal laceration and mortality, which can be curtailed by inhibiting PAD4 in a murine model of necrotizing enterocolitis [48,49]. Severe inflammation, intestinal necrosis, ulceration, as well as NET formation can be detected by light and fluorescence microscopy in the specimen of colonic resection from drug-crystal induced gastrointestinal complications [50]. In addition, NETs can produce numerous pro-inflammatory granular proteins, which may indirectly evoke the inflammation of local intestinal mucosa and eventually cause necroinflammation [51,52].

NETs also elicit a detrimental immune response leading to IBD through the production of neutrophil-driven granular proteins, including NE, MPO, histones, cathepsin G, and proteinase 3 (PR3), which can promote anti-neutrophil cytoplasmic auto-antibodies (ANCAs) production [53]. ANCAs are known to activate, complement, and cause endothelial damage [54]. In addition, they are also reported to generate a positive pro-inflammatory feedback loop by inducing NET formation [55]. Moreover, several studies have reported the presence of ANCA in serum of patients with IBD and/or in a murine model, which may be triggered by dendritic cells or MPO or PR3 [56,57]. In contrast, a recent review proposed beneficial roles of NETs in IBD [58]. Authors indicated favorable roles of circulating NETs in clearing damage associated molecular patterns based on previous studies [59,60,61]. However, none of these studies were conducted on NETs. In fact, they investigated the possible effects of bacterial DNA (bactDNA) and translocation of gut bacteria in IBD [59,60,61]. Since various gastrointestinal inflammation and diseases have been shown to be correlated with NET formation and NETs can be reciprocally related with an increased risk of developing intestinal malignancies, further studies are required to explore the axis of gastrointestinal inflammation and CRC in the context of NETs.

## 4. NETs in Cancer Progression and Metastasis

Over the years, NETs have been implicated in various types of cancer where they are involved in cancer growth or clearance, depending on cancer type, status of the immune system or tumor microenvironment [20,27]. Interestingly, higher levels of plasma NETs are present in cancer patients including lung, pancreatic, and bladder cancer [62]. In addition, NETs are found in lung tissues, serum, and sputum of lung cancer patients [63]. In the mice model, installation of cancer cells in the lung induces neutrophil recruitment and NET formation, suggesting that cancer cells itself can induce NET formation, perhaps by facilitating cancer cells adhesion and growth. Higher levels of NETs were observed in the liver metastases of patients with breast cancer, and serum NETs were identified as a predictive marker for the onset of liver metastases in patients with early-stage breast cancer [64]. In addition, it was revealed that CCDC25, a transmembrane protein, of breast cancer cells have the ability to sense distant NETs and navigate cancer cells to NETs [64]. An in vitro experiment has shown that NETs can induce breast cancer cells invasion and migration, and subsequent digestion of NETs by DNase I-coated nanoparticles reduces metastasis of breast cancer cells to lung in mice [65].

NET formation is observed during lung inflammation induced by smoke exposure or nasal instillation of lipopolysaccharide in animal models. NET associated proteases, neutrophil elastase, and matrix metalloproteinase 9, can cleave basal laminin and thus facilitate dormant cancer cells growth by activating integrin alpha-3 beta-1 signaling [66]. NETs can act as a trap to catch circulating cancer cells in the microvasculature of distant organs. In a murine model of sepsis, circulating lung carcinoma cells are reported to be trapped by NETs in the microvasculature of liver and cause gross metastatic burden after injection of tumor cells [22]. In addition, treatment with DNase or a neutrophil elastase inhibitor reduces cancer metastasis [22]. CD16^high^ and CD62^low^ neutrophil subpopulation possess higher NETs producing capacity and in head and neck squamous cell carcinoma patients, this subpopulation shows better survival [67]. Another study indicates that tumors can release granulocyte colony-stimulating factors into the bloodstream and promote the accumulation of intratumoral NETs and tumor growth by priming circulating neutrophils [68].

In gastric cancer (GC), the analysis of blood samples has revealed higher levels of NETs in patients with benign gastric disease than healthy controls [69]. The results indicate a better diagnostic value of NETs than carcinoembryonic antigen (CEA) and carbohydrate antigen 19-9 (CA19-9). These findings suggest the pivotal roles of NETs in the carcinogenesis of GC. In another study, it has been reported that low density neutrophils (LDN) from postoperative lavage generate a massive amount of NETs during the in vitro culture. In addition, the co-transfer of the peritoneal LDN with human gastric cancer cells enhance peritoneal metastasis in vivo [70].

## 5. COVID-19 and Cancer Progression

In recent times, due to the prevalence of ongoing worldwide COVID-19 pandemic, a considerable scientific interest has grown to determine the interdependence of COVID-19 in cancer progression and investigation of increased risk for potential and life-threatening outcomes from COVID-19 infections in the underlying medical condition of cancer. A recent systematic review involving 52 studies on COVID-19 and cancer has revealed that patients with cancer have a high probability of mortality due to the severe acute respiratory syndrome coronavirus 2 (SARS-CoV-2) [71]. One of the studies has indicated the presence of NETs in the lungs of autopsied COVID-19 patients [72], while another report has demonstrated the role of NETs in the initiation of immunothrombosis in COVID-19 patients [73]. It has been found that cancer patients could be more susceptible to SARS-CoV-2 infections due to the immunosuppression caused by chemotherapy or cancer itself. Furthermore, a meta-analysis has revealed that lung cancer and CRC patients are most susceptible to SARS-CoV-2 infection over other cancer types including breast, esophagus, bladder, pancreatic, and cervical cancer [74]. It is well documented that the expression of Angiotensin I-converting enzyme 2 (ACE2) on pulmonary epithelial cell plays the vital role in entering the virus into the body and interestingly, the expression of ACE2 is found to be higher in CRC tissues than matched normal tissues [75]. Furthermore, a higher expression of ACE2 has been observed in lung metastases from CRC than in normal lungs, suggesting that CRC patients are more vulnerable to SARS-CoV-2 infection [75]. The SARS-CoV-2 infection can be related to cancer pathogenesis as the progression of infection can alter the expression of the proteins involved in cell-cycle checkpoints, metabolism, and epigenetic regulation [76]. On the other hand, as hypoxia due to the SARS-CoV-2 infection can lead to poor oxygen supply to different organs, it can alter cancer cell metabolism and promote epithelial to mesenchymal transition [77]. For example, in breast cancer, hypoxia has been shown to promote gene expression involved in dormancy and drug resistance [78,79]. An alternative mechanism of cancer progression and metastasis during or after SARS-CoV-2 infection has been proposed, which indicates lowering of natural killer cells and T cells in the peripheral blood [80]. Taken together, it is our speculation that NET formation and lung inflammation by SARS-CoV-2 infection might trigger colon cancer cell migration and adhesion to inflamed organs.

## 6. NETs in Colorectal Cancer

Several studies have confirmed that patients with CRC can release elevated levels of NETs both in vivo and in vitro [25,81,82], which are mostly dispersed within the primary tumor sites and over the tumor boundary of CRC [26]. Although chemoradiotherapies and screening programs for early CRC detection are universal, about half the patients undergoing resection with therapeutic resolution tend to develop metastatic illness [83]. Accumulating evidences suggest that preoperative systemic inflammation could be involved in CRC recurrence following surgical resection [84]. In addition, several murine models and human observational studies demonstrated potential prognostic significance and association of NETs with CRC progression [84]. The recurrence and metastasis can be correlated with NETs production by perioperative systemic inflammation, such as sepsis or NETs production on the site of surgical wounds.

Several mechanisms have been proposed that can trigger NET formation in the CRC microenvironment (Figure 2). For instance, polyphosphate (polyP) expressed by CD68+ mast cells are shown to stimulate neutrophils to produce NETs in human colon carcinoma ex vivo [85]. KRAS, a small G protein of RAS family acts as a molecular switch in signal transduction pathways [86]. Activation of mutated KRAS regulates oncogenic malignant transformation and subsequent proliferation of cancer cells through activation of RAS/MAPK signaling pathway [82,87,88]. In 40–50% of CRC cases, KRAS mutations have been documented, hence, several studies have acknowledged it as a potential CRC prognostic and predictive marker [87,89]. Malignant cells can secrete exosomes to control the cellular microenvironment and KRAS mutant CRC cells have been shown to transfer mutant KRAS to neutrophils via exosomes [82]. The transfer of KRAS mutant protein by CRC cells induces neutrophil recruitment and subsequent NET formation by the upregulation of interleukin-8 (IL-8/CXCL8) both in vivo and in vitro [82]. The production of elevated levels of IL-8 and NET formation can act as a stimulator of CRC cell proliferation and can ultimately worsen the cancer condition [82]. IL-8 is known to recruit neutrophils and other myeloid leukocytes to converge at the site of infection via its receptors CXCR1 and CXCR2 [90,91]. IL-8 acts as a multifaceted chemotactic stimulus utilized by neoplastic cells to foster transmigration and angiogenesis concurrently [92,93]. Tumor cell-driven expression and excretion of IL-8 can also augment proliferation and survival of cancer cells by activating the autocrine system, promoting angiogenesis and infiltrating neutrophils into the malignant cells [90,91,94]. There is a clear correlation between IL-8 and NETosis in cancer progression and IL-8 mainly abets cancer progression, metastatic spread, and angiogenesis by directly priming the NET formation [82,90]. IL-8 along with its receptor CXCR2 is observed to provoke neutrophils towards the release of NETs by activating Src, ERK, and p38 signaling and the resultant released NETs can directly upregulate TLR9 pathways to stimulate cancer progression [95]. Moreover, IL-8 stimulates myeloid-derived suppressor cells via expressing CXCR1 and CXCR2 on their surfaces to extrude NETs which are shown to entrap cancerous cells [92]. Additionally, serum levels of IL-8 and its receptor CXCR2 are shown to highly upregulate in different phases of CRC compared to the normal samples. The secreted IL-8 profoundly stimulates human and murine CRC cell proliferation, incursion, migration, and amplifies angiogenesis around the tumor [91]. Moreover, IL-8-stimulated neutrophil-extruded NETs further advance the invasion and proliferation of CRC [91]. Lesions of CRC show a divergent surge of IL-8 where the upregulated IL-8 induces the activation of neutrophils and NET formation in CRC microenvironment [91,94].

It has been found that NET formation not only increases CRC cells proliferation but also stimulates the metastasis process [94,96]. NETs prime circulating tumor cells (CTC) adherence to hepatic or pulmonary endothelial surfaces [89,96,97] and thus, are involved in increased migratory pattern of CRC cells to the major critical regions of the body, such as liver, lung, and peritoneal cavity [25,96,98]. Surgical interventions for CRC are known to promote peritoneal carcinomatosis [99]. In fact, a recent study in mouse has shown that surgical trauma promotes colon cancer cells adhesion and growth in the peritoneal wall via CXCR2 signaling [100]. Population-based researches have reported that about 25–30% of CRC patients develop coetaneous liver metastasis and most of them show remarkably increased NET formation [25,96,101]. NETs show no cytotoxicity to the trapped CTCs in the liver but can raise their malignancy by enriching tumorous interleukin (IL-8) which in turn primes more NET formation, hence creating a positive loop for liver metastasis [94]. Furthermore, NET-associated carcinoembryonic antigen-related cell adhesion molecule 1 (CEACAM1) has been shown to stimulate the relocation of CRC cells to the liver both in vitro and in vivo [102].

A cohort of patients with analeptic liver resection for metastatic colorectal cancer has demonstrated the association of elevated postoperative NET formation with a lower survival rate [25]. In the same study, in a mouse model of liver metastasis and surgical stress, it has been shown that the inhibition of NET formation by DNase I reduces postoperative development of gross metastases [25]. Furthermore, several other studies demonstrate that diminishing of NETs using DNase reduces metastatic progression of CRC [82,94,101,102]. Table 1 summarizes the major outcomes of the studies describing NETs in CRC.

## 7. Therapeutic Potential of Targeting NETs in CRC

Accumulating evidence suggest that cancer pathology is correlated with NET formation. Until now, many therapeutic agents targeting NETs or NET formation or NET-associated components are successfully used in clinic and experimental diseases. It is our expectation that some of these agents could be useful to mitigate CRC progression and metastasis. For instance, several studies have successfully utilized DNase to degrade DNA backbone of NETs in different types of diseases including cystic fibrosis [104], colitis [39,105], IBD [47], CRC [82,94,101,102], breast cancer [65], and pancreatic cancer [106]. A recent meta-analysis of Cochrane Cystic Fibrosis and Genetic Disorders Group Trials Register reveals that the use of aerosolized recombinant DNase enzyme (Pulmozyme) improves the lung function with patients with cystic fibrosis compared to the placebo [107]. The DNase treatment effectively reduces viscoelasticity of DNA released by neutrophil with improved pulmonary function and well-being of patients. One advantage of using DNase is that the DNase-mediated NETs digestion does not hamper the physiological functions of neutrophils [28,108]. Furthermore, long before NETs were discovered several studies reported that DNA could act as a protective shield for harmful proteases [109,110]. Thus, elimination of DNA structure might neutralize the activity of NET-associated harmful proteases and enzymes and protect organs from damage and inflammatory events. In contrast, one recent study has reported the failure of DNase to eliminate NET associated harmful proteases or histones [108], which are known to cause tissue damage [111].

In addition to targeting NETs by DNase, alternative novel approaches targeting NETosis have been shown to reduce NET formation in several preclinical inflammatory disease models. For example, inhibition of ROS [112,113,114], PAD4 [39,115,116], NO/NOS [117], and Gasdermin D [118], have shown to reduce NETosis and/or disease progression. ROS-dependent formation of NET activates several sets of kinases (e.g., PKC, ERK, p. 38) via the activation of transcription factors (TFs), which in turn enables de-condensation of chromatin by PAD4 [28,35]. The inhibition of TFs does not hamper the immunological function of neutrophils but inhibits NET formation [112], thus it could be a suitable therapeutic approach to reduce NET-mediated CRC progression and metastasis. Moreover, the administration of PAD4 inhibitors (Cl-amidine and streptonigrin) showed promising therapeutic effects in DSS-induced UC [39]. Furthermore, the treatment with a high-affinity monoclonal antibody (infliximab) to TNF-α, diminishes the expression of PAD4 and TNF-α-driven NETosis in UC [46], suggesting that infliximab could also be used as a potential therapy targeting NETs in CRC. Gasdermin D (GSDMD) is a pore forming protein that facilitates NETs extrusion by puncturing granules which can be inhibited by employing a tiny size particle based on the pyrazolo-oxazepine scaffold that competently halts NETosis [118]. A NET-driven granular protein MPO has also been investigated as a target of anti-NETosis therapy. The inhibition of MPO has shown to lessen neutrophil recruitment and NETosis in the murine model of vasculitis and in vitro experiments [119,120]. Higher levels of inducible nitric oxide synthase (iNOS) expression and activity were detected in colon cancer specimens as compared to normal mucosa [121,122] and the use of NOS inhibitors together with 5-fluorouracil has shown enhanced reduction of colon cancer cells proliferation and migration [123]. Notably, the inhibition of NOS has shown to reduce nitric oxide (NO)-mediated NET formation in vitro [117], suggesting that the NOS inhibitor could also be used as a potential therapy in CRC management. Interestingly, metformin, a well-known clinically established antidiabetic drug, has been reported to reduce NET formation via the inhibition of PKC-NADPH-oxidase pathway [34,124]. In addition, the nanocarrier based combination treatment, such as Oshadi D (DNase in an Oshadi carrier) and Oshadi R (RNase in an Oshadi carrier), has shown promising anti-cancer effects in phase II clinical trial (ClinicalTrials.gov Identifier: NCT02462265), possibly by targeting NETs. Autophagy inhibitor, hydroxychloroquine, was also shown to reduce NET-mediated hepatic ischemia/reperfusion (I/R) injury by inhibiting PAD4 and Rac2 expression [125]. It should be noted that neutrophil plays a key role in innate immune response, therefore, targeting NETs or mechanisms of NETosis or NET-driven products should be implemented in such a way that the intervention should insulate the fundamental physiological function of neutrophils. Table 2 summarizes studies and therapeutics used to target NETs in various diseases. 

## 8. Conclusions

Despite significant preclinical and clinical research on CRC, mortality remains high when cancer progresses to multiple organs. This could be due to poor understanding of pathological mechanism of CRC in the context of inflammation and NET formation. Inflammation plays a profound role throughout the whole process of carcinogenesis staring from initiation of primary tumor to metastasis. Moreover, neutrophils and neutrophil-released products have been implicated in various types of cancer progression and metastasis. Although some studies showed anti-tumor roles of neutrophils, higher levels of neutrophils numbers in primary cancer and pre-metastatic organs were shown to associate with cancer progression and metastasis. The formation of NETs is an indispensable mechanism of host response where neutrophils kill and trap pathogens. However, in various cancer types, NETs were found to promote cancer cells growth and metastasis by trapping circulating cancer cells in distant inflamed organs. Although, a small number of studies investigated possible roles of NETs in CRC, increasingly a robust body of evidence indicated that NETs might play a significant role in the pathophysiology of colon cancer. In recent times, several experimental studies targeting NETs and NET-associated proteins showed promising results in mitigating disease progression and metastasis. Thus, it could be suggested that in conjunction with surgery and adjuvant chemotherapy, new treatment strategies in order to prevent NET-mediated CRC progression and metastasis would be a promising approach in the clinical settings. It should be noted that recombinant human DNase I has been used for patients with cystic fibrosis and systemic lupus erythematosus, respectively and appears to be safe and effective. Therefore, targeting NETs or NET formation could provide a promising strategy to inhibit both progression and metastasis of CRC.

## Figures and Tables

**Figure 1 ijms-22-07260-f001:**
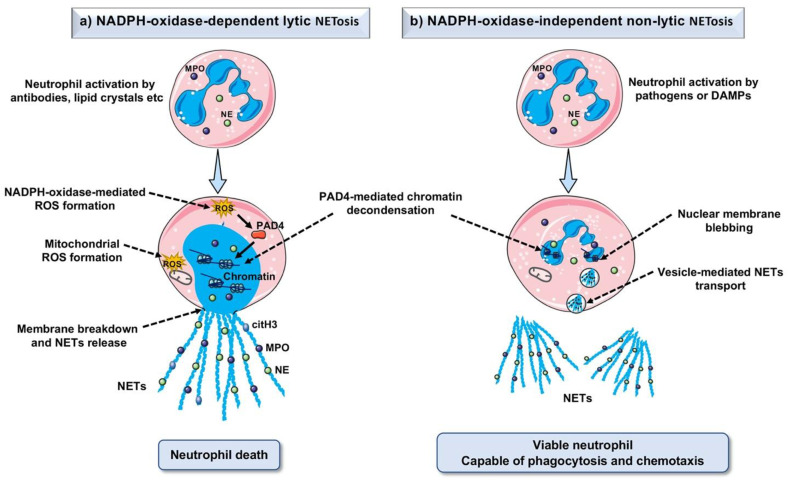
Mechanisms of NET formation. (**a**) NADPH-oxidase (NOX)-dependent lytic NET formation: After activation, neutrophils produce reactive oxygen species (ROS) via the NADPH-oxidase complex. ROS activates or upregulates protein-arginine deaminase type 4 (PAD4, which promotes citrullination of histones and subsequent chromatin de-condensation). Myeloperoxidase (MPO) helps translocate neutrophil elastase (NE) into the nucleus, which leads to further chromatin de-condensation, finally the nuclear membrane is disrupted, and NETs decorated with granular and cytosolic proteins are released in extracellular space. Neutrophils die after NET formation. (**b**) NADPH-oxidase (NOX)-independent non-lytic NET formation: After neutrophils activation by pathogens or DAMPs, PAD4 promotes chromatin de-condensation. NETs decorated with granular and cytosolic proteins are released outside via vesicular transport without plasma membrane disruption. After the release of NETs, neutrophils remain viable, and capable of phagocytosis and chemotaxis.

**Figure 2 ijms-22-07260-f002:**
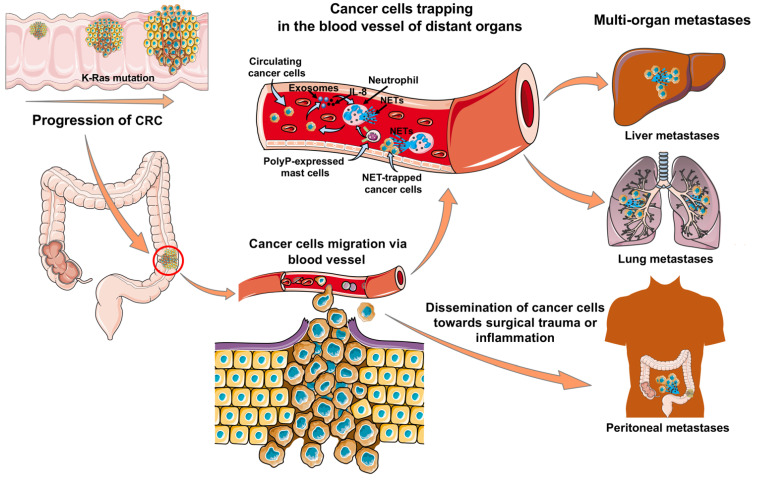
NET-mediated CRC progression and metastasis. KRAS mutation induces colon cancer cells proliferation and migration through the blood vessel. IL-8 secretion by circulating cancer cells or transfer of mutant KRAS to neutrophils via exosomes provoke NET formation in blood vessel of distant organs. Mast cells expressing polyphosphates (polyP) stimulate neutrophils and produce NETs. NETs trap cancer cells in distant organs such as in the liver and lung and thus, help metastatic spread. Spontaneous dissemination of colon cancer cells towards peritoneal wounds or inflammation and subsequent attachment to the inflamed peritoneum promote peritoneal metastases.

**Table 1 ijms-22-07260-t001:** Summary of the included studies of NETs in CRC.

Study Design	Animal Models	Cell Type	Major Outcomes/Findings	Authors
Both in vivo and in vitro	Mouse	Human colon carcinoma cell line (HT-29), murine colon carcinoma	NET-associated protein CEACAM1 is an inducer of metastatic progression of CRC and blocking of NETs significantly reduce CRC cell adhesion, migration, and metastasis in murine model.	Rayes et al. [102]
Both in vivo and in vitro	Mouse	DKs-8 (WT allele) cells, DKO-1 (KRAS mutant) cells	Exosomes from KRAS mutant CRC increase IL-8 production and provoke NET formation. Released NETs increase CRC cells growth both in vivo and in vitro.	Shang et al. [82]
In vivo, in vitro, and ex vivo	Human and Mouse	Human hepatocellular carcinoma, human cell line HT29, and mice cell line MC38	NETs raised colorectal malignancy by enriching tumorous interleukin IL-8, which in turn induce more NET production by creating a positive loop along with advancing CRC-driven liver metastasis. Digestion of NETs by DNase I reduced liver metastasis.	Yang et al. [94]
Both in vivo and in vitro	Mouse	Human hepatoma cell line HepG2, murine colon carcinoma MC38	Neutrophil infiltration and NET formation reduced by adeno-associated virus (AAV) based DNase I gene therapy and reduced liver metastasis in a mouse model of CRC liver metastasis.	Xia et al. [101]
Both in vivo and in vitro	Mouse	Murine Lewis Lung carcinoma cell subline H59, Murine colon carcinoma cell line MC38	Primary colon cancer cells provoked NETs generation that prime adhesion of CTCs to the liver and degradation of NETs decreased CRC cell adhesion and spontaneous metastasis to the liver and lung.	Rayes et al. [96]
Both in vivo and ex vivo	Human and Mouse	Murine colorectal (MC38) cells, HCT116, Hepa1-6, and Huh7 cell lines	Patients undergoing curative resection with colorectal metastases to the liver showed an elevated level of NET formation. Increased citrullinated histones and circulating MPO-DNA levels were related to poor survival of CRC patients.	Yazdani et al. [98]
Ex vivo	Human	CRC cells	CD68+ mast cells expressed polyphosphates (PolyP) in colorectal adenomas and/or carcinomas and suggested that CD68+ PolyP expressing mast cells could be used as prognostic marker.	Arelaki et al. [85]
Ex vivo	Human	/	Systemic neutrophils isolated from the CRC patients showed higher levels of NETs producing ability than healthy controls in vitro. In vitro increased NET production is correlated with patients’ major complications than minor complications.	Richardson et al. [81]
Ex vivo	Human	/	Neutrophils isolated from patients undergoing resectional surgery for CRC showed lower NET forming ability in vitro than preoperative neutrophils.	Richardson et al. [103]
In vitro and Ex vivo	Human	Human acute myeloid leukemia (AML) cells, Caco-2 cells	Confirmed presence of NETs within the primary tumor sites of CRC and gradually dispersed to the tumor boundary, particularly to nearby metastatic lymph nodes.	Arelaki et al. [26]
In vivo, in vitro, and ex vivo	Human and Mouse	MC38 and Luciferase-expressing MC38 cells (MC38/Luc)	Increased postoperative NETs generation after curative liver resection of colorectal metastasis patients. NETs further fuel the metastasis condition and reduce more than 4-fold disease free survival.	Tohme et al. [25]

**Table 2 ijms-22-07260-t002:** List of studies targeting NETs in various diseases.

Studies	Therapeutic Agents	Targets	Mechanism of Action	Major Findings	Disease/Model
Yang et al. [94]	DNase 1	DNA backbone of NETs	Digestion of NETs	Diminished colorectal cancer liver metastasis.	CRC (in vivo)
Xia et al. [101]	Adeno-associated virus (AAV)-mediated gene transfer of DNase I	DNA backbone of NETs	Digestion of NETs	Reduced liver metastasis in a mouse model of CRC liver metastasis.	CRC (in vivo)
Rayes et al. [102]	DNase	DNA backbone of NETs	Digestion of NETs	Inhibited CRC cell adhesion and migration in vitro. Reduced liver metastasis of CRC cells.	CRC (in vivo and in vitro)
	CEACAM1 blocking antibody	NET-associated CEACAM1	Blocking of CEACAM1 on NETs		
Shang et al. [82]	DNase	DNA backbone of NETs	Degradation of NETs	Reduced KRAS mutant exosome-induced CRC cells adhesion.	CRC (in vitro)
Shah et al. [104]	DNase	DNA backbone of NETs	Degradation of NETs	Reduced viscoelasticity of sputum and improved pulmonary function.	Cystic fibrosis (clinical trials)
Li et al. [47]	DNase	DNA backbone of NETs	Degradation of NETs	Lessened cytokine levels, attenuated thrombus formation and activation of platelet.	DSS-induced colitis (in vivo)
Park et al. [65]	DNase	DNA backbone of NETs	Degradation of NETs	Inhibited NET-induced invasion and migration of breast cancer cells in vitro. Reduced breast cancer cells metastasis to lung.	Breast cancer (in vitro and in vivo)
Xiao et al. [126]	AZD7986 (inhibitor of Cathepsin C)	Cathepsin C	Inhibit CTSC-PR3-IL-1β axis mediated reactive oxygen species production	Reduced lung metastasis of breast cancer in a mouse model.	Breast cancer (in vivo)
Wen et al. [106]	DNase	Extracellular DNA (exDNA)	Degradation of exDNA	Suppressed metastasis of pancreatic cancer cells in an orthotopic xenograft model.	Pancreatic cancer (in vivo)
Sollberger et al. [118]	Gasdermin D Inhibitor (LDC7559)	Pore-forming protein Gasdermin D (GSDMD)	LDC7559 binds to GSDMD and prevents pore formation in granule membrane	Decreased phorbol 12-myristate 13-acetate (PMA)-induced NET formation.	In vitro
Khan et al. [112]	Actinomycin D and Topoisomerase I inhibitor	Promoter region of DNA	Inhibit protein transcription initiation	Blocking of transcription suppresses NETosis without affecting ROS generation.	In vitro
Lood et al. [113]	MitoTEMPO	ROS	MitoTEMPO scavenge mitochondrial superoxide	Mitochondrial ROS inhibition reduced NET formation and systemic lupus erythematosus (SLE) disease severity.	SLE (in vivo and in vitro)
	Apocynin	ROS	Block superoxide production	Reduced PMA-induced NET formation.	
	VAS2870	ROS	Inhibit NADPH-oxidase (NOX)	Reduced PMA-induced NET formation.	
Van Avondt et al. [114]	Diphenyleneiodonium (DPI)	NADPH-oxidase	Inhibit ROS generation	Reduced PMA-induced NET formation.	In vitro
Knight et al. [115]	Cl-amidine and BB-Cl-amidine	PAD4	Inhibit PAD4	PAD inhibition diminished NET formation and showed protection against lupus-related damage to vasculature, kidney in murine lupus model.	SLE (in vivo)
Maronek et al. [39]	Cl-amidine and Streptonigrin	PAD4	Inhibit PAD4	Reduced plasma level of ecDNA but could not lessened total UC condition in mice.	DSS-induced UC (in vivo)
Dinallo et al. [46]	Infliximab (anti-TNF-α antibody)	TNF-α	Block TNF-α	Reduced PAD4 expression and TNF-α-driven NETosis.	UC (in vivo)
Zheng et al. [119]	PF-1355	MPO	Inhibition of MPO	Decreased neutrophil recruitment and NETosis.	In vitro
Parker et al. [120]	ABAH (4-aminobenzoic acid hydrazide)	MPO	Inhibition of MPO	Reduced PMA-induced NET formation.	In vitro
	TX1 (3-isobutyl-2-thioxo-7H-purine-6-one)				
Smith et al. [127]	Chloroquine			Reduced LPS-induced NET formation.	In vitro
Fuchs et al. [128]	Heparin	Histones	Remove histones from NETs and destabilize NETs	Reduced NET formation.	In vitro
Manda-Handzlik et al. [129]	Apocynin and DPI	NADPH-oxidase	Inhibit NADPH-oxidase activity	Reduced S-nitroso-N-acetyl-D,L-penicillamine (SNAP)-induced NET formation.	In vitro
	N-acetylcysteine (NAC)	ROS scavenger	Interfere with the levels of hydrogen peroxide and hydroxyl radical	Inhibited NO-dependent NETosis	
Li et al. [117]	SMT	Inducible NO synthase (iNOS)	Block NO synthesis	Inhibited NO-mediated NET formation	In vitro
	L-NAME	Endothelial NO synthase (eNOS)			
	L-NMMA	Total NOS			
Wang et al. [34]	Metformin (antidiabetic)		Inhibit mitochondrial respiratory chain complex I and NADPH-oxidase activity, thus decrease ROS production	Reduced PMA-induced NET formation	In vitro
Menegazzo et al. [124]	Metformin (antidiabetic)		Inhibit membrane translocation of PKC-βII and activation of NADPH-oxidase	Reduced NET components elastase, proteinase-3, histones, and double strand DNA in the plasma of pre-diabetes.	Pre-diabetes (in vivo and in vitro)
Zhang et al. [125]	Hydroxychloroquine (autophagy inhibitor)		Inhibit PAD4 and Rac2 expressions by blocking TLR9	Reduced hepatic ischemia/reperfusion (I/R) injury by inhibiting NET formation.	Hepatic I/R injury (in vivo and in vitro)
Phase II clinical trial (NCT02462265, https://clinicaltrials.gov) Accessed on 30 June 2021	Oshadi D (DNase) and Oshadi R (RNase)	DNA and RNA		Showed antitumor activity and a good safety profile in leukemia patients.	Acute myeloid leukemia or acute lymphoid leukemia

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
