# Peer review of "Neutrophil Extracellular Traps in Colorectal Cancer Progression and Metastasis"

_ijms, 2021, doi:10.3390/ijms22147260_

Round 1

Reviewer 1 Report

The manuscript entitled "Neutrophil extracellular traps in colorectal cancer progression and metastasis" highlights the main findings that support the role of NET in CRC. The material is well-written, clear and easy to follow, and up-to-date. I suggest just some minor modifications before publication:

- in table I, I would recommend the removal of the 2 lines that are linked to previously published reviews and insert the information into the text; for a better understanding, it is recommended to not mix actual results with summarized literature;

- the insertion of a table that summarizes the current therapy available presented in section 7 will be appreciated for creating a clear picture for the reader, since this section that presents targeting NET as a promising pharmacological strategy is a feature of great importance in this review; at least a graphical representation should be added in this section;

- I would recommend the expansion of the conclusions since I would not consider that in this form it reflects the key information discussed in the review;

- Please read carefully the material for English minor check such as incorrect article usage, missing determinants and so on;

Author Response

First, we would like to thank the reviewer for constructive comments that allowed us to improve the manuscript. 

Response to Reviewer 1.

In table I, I would recommend the removal of the 2 lines that are linked to previously published reviews and insert the information into the text; for a better understanding, it is recommended to not mix actual results with summarized literature;

Answer: We appreciate the useful comments of the reviewer. We now deleted indicated review articles from the Table 1 and added the relevant information in the main text of the revised manuscript. (lines 217-221, marked red in the revised manuscript)

The insertion of a table that summarizes the current therapy available presented in section 7 will be appreciated for creating a clear picture for the reader, since this section that presents targeting NET as a promising pharmacological strategy is a feature of great importance in this review; at least a graphical representation should be added in this section;

 Answer: We appreciate the suggestion of this reviewer. We now added Table 2 summarizing studies targeting NETs and potential therapeutics. (page 11-14, marked red in the revised manuscript)

In addition, information regarding therapeutic agent ‘Avacopan’ is now deleted from the manuscript since the article did not discussed NETs.

I would recommend the expansion of the conclusions since I would not consider that in this form it reflects the key information discussed in the review;

Answer: The conclusion is now expanded with key information. (lines 346-359, marked red in the revised manuscript)

Reviewer 2 Report

In the review entitled “Neutrophil extracellular traps in colorectal cancer progression and metastasis” the authors have summarized the current understanding of the role of Neutrophil extracellular trap in the pathology of colorectal cancer (CRC). They further discussed the regulatory mechanism of NETosis and emphasize it as a potential new therapeutic target for colorectal cancer management. The review is very comprehensive, nicely written, and adequately covers the current literature in the field. The figures and tables are accurately represented.

Some concerns raised are shown below.

The review provides some background on the role of NETs in colorectal cancer with recent references. I recommend citing the study suggesting the role of transmembrane protein CCDC25 in cancer metastasis and cancer patients’ poor prognosis (PMID: 32528174).

Since several drugs including metformin (antidiabetic), hydroxychloroquine (autophagy inhibitor), and Oshadi D (DNase in an Oshadi carrier) and Oshadi R (RNase in an Oshadi carrier) demonstrate anti-NET activity, the authors should discuss the feasibility of these strategies in colorectal cancer management.  

Nitric oxide (NO) plays a crucial role in NETosis, and ubiquitous expression of inducible nitric oxide synthase associated with inflammatory colorectal cancer cell growth and metastasis. The authors should discuss the link between NO/NOS targeting therapies and NETs inhibition for CRC management.

Author Response

First, we would like to thank the reviewer for constructive comments that allowed us to improve the manuscript. 

Response to Reviewer 2.

The review provides some background on the role of NETs in colorectal cancer with recent references. I recommend citing the study suggesting the role of transmembrane protein CCDC25 in cancer metastasis and cancer patients’ poor prognosis (PMID: 32528174).

Answer: As suggested by the reviewer, we now cited the article regarding transmembrane protein CCDC25 and discussed key information in lines 154-158 (marked red).   

Since several drugs including metformin (antidiabetic), hydroxychloroquine (autophagy inhibitor), and Oshadi D (DNase in an Oshadi carrier) and Oshadi R (RNase in an Oshadi carrier) demonstrate anti-NET activity, the authors should discuss the feasibility of these strategies in colorectal cancer management.

 Answer: We appreciate the reviewer valuable suggestion. We now discussed metformin (antidiabetic), hydroxychloroquine (autophagy inhibitor), and Oshadi D (DNase in an Oshadi carrier) and Oshadi R (RNase in an Oshadi carrier) (lines 337-343) and listed as potential therapeutic agents in the new table (Table 2).

 In addition, information regarding therapeutic agent ‘Avacopan’ is now deleted from the manuscript since the article did not discussed NETs.

Nitric oxide (NO) plays a crucial role in NETosis, and ubiquitous expression of inducible nitric oxide synthase associated with inflammatory colorectal cancer cell growth and metastasis. The authors should discuss the link between NO/NOS targeting therapies and NETs inhibition for CRC management.

Answer: The link between NO/NOS and NETs in the context of CRC management is now discussed in the new version of our manuscript. (lines 337-340, marked red)